# Preventive Measures against Pandemics from the Beginning of Civilization to Nowadays—How Everything Has Remained the Same over the Millennia

**DOI:** 10.3390/jcm11071960

**Published:** 2022-04-01

**Authors:** Laura Vitiello, Sara Ilari, Luigi Sansone, Manuel Belli, Mario Cristina, Federica Marcolongo, Carlo Tomino, Lucia Gatta, Vincenzo Mollace, Stefano Bonassi, Carolina Muscoli, Patrizia Russo

**Affiliations:** 1Laboratory of Flow Cytometry, IRCCS San Raffaele Roma, 00166 Rome, Italy; laura.vitiello@sanraffaele.it; 2Institute of Research for Food Safety & Health (IRC_FSH), Department of Health Sciences, University ‘Magna Graecia’ of Catanzaro, 88100 Catanzaro, Italy; sara.ilari@hotmail.it (S.I.); mollace@libero.it (V.M.); muscoli@unicz.it (C.M.); 3MEBIC Consortium, IRCCS San Raffaele Roma, 00166 Rome, Italy; luigi.sansone@sanraffaele.it (L.S.); manuel.belli@sanraffaele.it (M.B.); mario.cristina@sanraffaele.it (M.C.); 4Clinical and Molecular Epidemiology Unit, IRCCS San Raffaele Roma, 00163 Rome, Italy; federica.marcolongo@sanraffaele.it (F.M.); stefano.bonassi@sanraffaele.it (S.B.); 5Scientific Direction, IRCCS San Raffaele Roma, 00166 Rome, Italy; carlo.tomino@sanraffaele.it (C.T.); lucia.gatta@sanraffaele.it (L.G.); 6Department of Human Sciences and Quality of Life Promotion, San Raffaele University, 00166 Rome, Italy

**Keywords:** history of plague, preventive measures, COVID-19, SARS-CoV-2, vaccines

## Abstract

As of 27 March 2022, the β-coronavirus severe acute respiratory syndrome coronavirus 2 (SARS-CoV-2) has infected more than 487 million individuals worldwide, causing more than 6.14 million deaths. SARS-CoV-2 spreads through close contact, causing the coronavirus disease 2019 (COVID-19); thus, emergency lockdowns have been implemented worldwide to avoid its spread. COVID-19 is not the first infectious disease that humankind has had to face during its history. Indeed, humans have recurrently been threatened by several emerging pathogens that killed a substantial fraction of the population. Historical sources document that as early as between the 10th and the 6th centuries BCE, the authorities prescribed physical–social isolation, physical distancing, and quarantine of the infected subjects until the end of the disease, measures that strongly resemble containment measures taken nowadays. In this review, we show a historical and literary overview of different epidemic diseases and how the recommendations in the pre-vaccine era were, and still are, effective in containing the contagion.

## 1. Introduction

A novel single-stranded RNA virus from the β-Coronaviridae family named severe acute respiratory syndrome coronavirus 2 (SARS-CoV-2), initially recognized as the viral cause of a cluster of pneumonia “originally unknown”, emerged firstly in China, spread rapidly worldwide, and led to a global pandemic called “coronavirus disease 2019” (COVID-19). β-Coronaviruses are members of a large family of RNA viruses. One-third of the RNA sequence encodes four core structural proteins: spike (S), membrane-associated envelope (E), membrane (M), and nucleocapsid (N) proteins [1].

COVID-19 is characterized by severe upper respiratory tract infections, respiratory distress, and frequent need for hospitalization. Since the outbreak, COVID-19 has killed more than 6.12 million patients worldwide as of March 27th 2022. The clinical spectrum of COVID-19 ranges from asymptomatic or pauci-symptomatic to severe symptomatic with a large range of clinical manifestations such as cough, fever, myalgia, gastrointestinal symptoms, and anosmia. Admission to intensive care units (ICUs), mechanical ventilation, and mortality are common, especially in subjects with compromised immune systems, in subjects affected by underlying chronic diseases, or in the elderly [2]. The rapid deterioration of clinical conditions has been described as a typical feature of SARS-CoV-2 infection, driven by the pathogen–host interaction. The angiotensin-converting enzyme 2 (ACE2) is the only confirmed SARS-CoV-2 entry receptor [3]. SARS-CoV-2 and ACE2 binding allows the virus to enter a cell through the spike (S) proteins that work in concert with the host cell serine protease transmembrane protease serine 2 (TMPRSS2). TMPRSS2 cleaves the spike protein of SARS-CoV-2, facilitating membrane fusion. ACE2 relies on the renin–angiotensin system (RAS) molecular pathway; ACE2 is a crucial counter-regulatory enzyme to ACE by the breakdown of angiotensin II that is involved in blood pressure regulation and electrolyte homeostasis [4]. ACE2 is expressed in multiple tissues, and thus the existence of multiorgan complications/failure is not surprising [5]. Recently, clinicians started to report prolonged sequelae of symptoms in the post-acute phase of COVID-19, which may be represented in three categories:(1)Residual symptoms continuing after acute infection recovery;(2)Organ dysfunction continuing after initial recovery;(3)New symptoms/syndromes developing after initial asymptomatic or mild infection [6].

As a consequence of the long duration of pandemics and the large number of patients that survived the disease, the priority for National Health Systems is progressively shifting towards mid- and long-term effects of COVID-19. Frail and prefrail subjects who recovered after hospital admission represent an excellent model for studying this new scenario. With the COVID-19 pandemic, consideration of frailty hallmarks in dismissing infected older patients is critical to identify those subjects at higher risk of functional decline that may fall into accelerated aging processes. In this contest, there is the vital role of rehabilitation programs for the coming years and the urgent need to develop strategies to assist COVID-19 survivors.

### Communicable Diseases

Communicable diseases, also known as transmissible diseases or infectious diseases, are illnesses that spread, directly or indirectly, from one organism to another through the transfer of a pathogen such as viruses, bacteria, fungi, protozoa, multicellular parasites, and aberrant proteins known as prions. They have been present for all of human history but became important with the rising of agrarian life 10,000 years ago when the demographic expansion and technological innovations made the transmission of infectious diseases more possible. The zoonotic transmission of pathogens from animals to humans is the key mechanism in the process of emerging infections, and the probability of cross-species transmission is enhanced by increasing interactions between humans and animals [7]. According to Wolfe et al. [7], the passage of a pathogen from a different species to humans is characterized by five steps: (1)The pathogen infects only animals under natural conditions;(2)The pathogen evolves to be transmitted to humans without continuous human-to-human transmission;(3)The pathogen undergoes secondary transmission to humans;(4)The disease exists in animals but different secondary human-to-human transmission occurs without the involvement of animal hosts;(5)The disease occurs exclusively in humans.

The risk of zoonotic transmission is related to the animal species harboring the pathogen, the nature of human–animal interaction, and the frequency of these interactions.

One of the first pieces of evidence of the insurgence of an epidemic is derived from molecular studies that unambiguously identified the presence of DNA from *Yersinia pestis* (*Y. pestis*), the causative agent of plague, in different individuals at Fralsegarden in Gokhem parish, Falbygden, western Sweden, dated to 4900 BCE [8]. Interestingly, the analysis of both the archaeological context and the human genomes supports the notion that the rise and the spread of the above plague were related to lifestyle (high population densities in close contact with animals), population growth, and expanding trade networks. It has been proposed that the *Y. pestis* plague may have contributed to the Neolithic decline [8].

## 2. History of “Plagues”

The written texts and artworks of past times are useful as they provide information about the cultural contexts and enhance the understanding of the spread of communicable diseases in the past, centuries before the modern scientific literature. Literary plagues involve different forms, from true-to-life to completely imaginary. The term “plague” derives from the Latin plaga, meaning “blow or wound” [9]. Although plague refers to infection by an extremely contagious Gram-negative bacterium, *Y. pestis* (formerly called *Pasteurella pestis*), the word “plague” is used not only as a generic term for an epidemic or pandemic but also as a metaphor for a wide range of calamities. Interestingly, the first book of Western literature, Homer’s Iliad, starts with the story of a plague that strikes the Greek army at Troy (Table 1) [10,11,12,13,14,15,16,17,18,19,20,21,22,23,24,25,26,27,28]. 

It is important to remember that no medicine or doctor had been able to cure or prevent infectious diseases for millennia, and the only way to stay safe was to avoid contact with infected people and contaminated objects. The great terror of epidemics was also fueled by the belief in their supernatural origin, as they were often believed to be caused by irate gods. In the Bible (i.e., Exodus (Hebrew: יציאת מצרים) 9:14, Numbers (Hebrew: בְּמִדְבַּר) 11:33, Former Prophets 1 Samuel (Hebrew: נביאים ראשונים, שְׁמוּאֵל]) 4:8, Psalms (Hebrew: תְּהִלִּים) 89:23, the Latter Prophets Isaiah (Hebrew: נביאים אחרונים; ספר ישעיהו) 9:13), the plague was regarded as a divine punishment for sins, and hence the frightening description of its diffusion was a warning to behave morally (reviewed also by Freemon [29]). This supposed causal relationship between plague and sin is also present in Greek literary texts, such as Homer’s Iliad and Sophocles’ King Oedipus (429 BCE) (Table 1). This attitude changed completely with Thucydides (460–395 BCE), a Greek historian, who in his History of the Peloponnesian War, refused a supernatural origin of the disease (the Plague of Athens), describing it with a scientific approach, reporting the origin of the plague, the symptoms, the incapacity of doctors to cure it, and the uncontrolled fear of contagion among the public (Table 1). The plague originated in early May 430 BCE, with a second wave in the summer of 428 BCE and a third in the winter of 427–426 BCE, and spared no segment of the population, including the statesman Pericles and Thucydides himself. The Plague of Athens killed between 75,000 and 100,000 people, ~25% of the population of Athens (429 BCE) (Table 1 and Table 2 and Figure 1) [16,30].

The Roman historian Titus Livius, known as Livy in English (Table 1), writing many centuries later, reported that epidemics occurred also in Rome in 433 and 438 BCE, suggesting the same origin of the disease. DNA analysis of skeletal remains recovered in 2001 in a mass burial pit in Greece and dating back to the plague years revealed ancient microbial typhoid (*Salmonella enterica* serovar *Typhi*), supporting the hypothesis that the Plague of Athens was a typhoid disease [32].

Figure 1 reports the emergence of the most important pandemics in the world until today with a possible estimation of the number of casualties, whereas Figure 2 reports the origin of the pandemics, considering their onset for the first time.

Centuries later, between 165 and 180 CE, the Roman empire was challenged by the Antonine Plague. It is supposed that the disease was brought by armies from what is currently Iraq and eventually spread into the entire Roman empire, which comprised Central and Southern Europe. It is estimated that the diseases killed about 5 million people [20]. The symptoms were described by Galen and suggested that it was probably smallpox, as its presentation was characterized by rashes, hemorrhagic pustules, bloody diarrhea, fever, and sometimes hemoptysis [33].

The Justinian Plague (starting in 541 CE with subsequent outbreaks until 750–1000 CE) is the first confirmed pandemic plague caused by *Y. pestis* based on the analysis of the teeth of people buried at that time [22]. Accordingly, the reported symptoms were those distinctive of *Y. pestis* infection: fever, cough, and dyspnea in pneumonic plague and groin or axillary buboes in bubonic plague. The transmission was mediated from infected rats to humans through flea bites, but there was also human–human transmission. It has been estimated that the Plague of Justinian killed 60% of people in the Mediterranean area [34].

The Justinian Plague may have contributed to the end of the Roman empire, characterizing the transition from the Classical to the Medieval era [22]. It has been shown that the strain of *Y. pestis* linked with the Plague of Justinian is different from those associated with later human plague pandemics, and it seems that this strain is extinct or poorly present in wild rodent reservoirs [22]. 

The first scientific description of the etiology of plague and of the possible way of spreading of the disease did not become available until the 19th century. In 1894, in Hong Kong, Alexander Yersin (1863–1943) isolated in culture and identified the causative bacterial agent (i.e., *Yersinia pestis*) [35]. After examination of the lymph glands of dead rats, he found the same bacteria described previously in human tissues; consequently, he made the causal connection between rat mortality and human epidemics. Surprisingly, the earliest recorded role of mice in plague is in the Bible that ascribed the pestilence among the Philistines to “the mice that marred the land” (1 שְׁמוּאֵל Samuel. 6:4–18). 

For the next centuries, after the different outbreaks of Justinian’s plague, different epidemics occurred frequently. The next great pandemic plague was the dreaded “Black Death” of Europe in the 14th century (Figure 1 and Figure 2 and Table 1). 

The Black Death, also known as the Black Plague, spread from the Caspian Sea by the town of Caffa, now Feodosia, a Genoese colony, to almost all European countries, causing the death of one-third of the European population [36] over the next few years, and persisted in Europe until 1750. Following traditional beliefs promoted by Hippocrates and Galen, the poisoned air (“miasma”) was believed to be the causative agent of the epidemics; as already mentioned, an association between the plague and dead rats was not observed until the beginning of the 20th century [30]. The name Black Death derives from a typical symptom of the disease, called acral necrosis, of black color due to subdermal hemorrhages. Although there was a great debate on the etiology of this disease, DNA and protein signatures specific to *Y. pestis* were found in human skeletons from mass graves associated archaeologically with the Black Death throughout Europe, confirming it as the causative agent. Furthermore, the analysis of 17 single nucleotide polymorphisms (SNPs) and the absence of a deletion in the glpD gene (aerobic glycerol-3-phosphate dehydrogenase) identified two previously unknown but related clades of *Y. pestis* associated with distinct medieval mass graves, suggesting that the plague arrived in Europe at different times, through distinct routes [37]. The third wave of *Y. pestis* started in 1772 in Yunnan Province, Southwest China. In this case, Europe was not significantly affected, but some cases were reported in Malta in June 1945 [35,38]. 

A great nightmare and terror worldwide was represented by smallpox, which was a highly contagious and lethal disease with remarkably high death rates until its global eradication, declared by the WHO on 8 May 1980. The causative agent of smallpox is the variola virus (VARV), a member of the genus *Orthopoxvirus*, family *Poxviridae*, and subfamily *Chordopoxvirina*. Analysis of viruses’ DNA collected from different individuals living both in Eurasia and in the Americas between 200 and 150 years ago revealed that their DNA resembled modern variola. The young pharaoh Rameses V in the 12th century BCE was probably killed by VARV [39]. The etiological cause of the Antonine Plague, which killed the emperor himself, was probably VARV. Queen Elizabeth I of England, at the age of 29 years in October 1562, survived smallpox, which left her without hair and with permanent disfiguring facial scars [40]. In 1694, VARV killed Queen Mary of England; six years later, it killed her son, the Duke of Gloucestershire, and then it went on to kill Emperor Joseph I of Austria (17 April 1711), King Luis I of Spain (31 August 1724), Tsar Peter II of Russia (30 January 1730), Queen Ulrika Eleonora of Sweden (24 November 1741), and the French King Louis XV (10 May 1774) [41]. During the American Revolutionary War (1775–1783), one of the greatest threats to the army was smallpox. In 1775, General George Washington recognized smallpox as a very serious problem for his army, especially after the smallpox outbreak in the city of Boston in the winter of 1775 [42].

Table 2 shows the scientific aspect of the history of infectious diseases [43,44,45,46,47,48,49,50,51,52,53,54,55,56,57,58,59,60,61,62,63,64,65,66,67,68,69,70,71,72,73,74,75,76,77,78,79,80], and Table 3 shows infectious diseases studied using molecular techniques, including metagenomics. Different reviews as well as two important books explore this field [81,82,83,84,85,86]. Moreover, Spyrou et al. [87] reviewed the ancient pathogen genomic data recovered from archaeological or historical specimens, including the method of retrieval. DNA analysis from archaeological sites has been a powerful tool for deciphering the steps of the evolution of pathogens [81]. Combining the data obtained from molecular analysis with disease modeling and the genetic history of the human population, together with the information offered by the archaeological, historical, and palaeopathological records, is helpful to define a comprehensive picture of host–pathogen interactions during centuries. Artistic and literary works, on the other hand, are helpful in understanding what kind of measures were taken to deal with pandemic emergencies in ancient times.

**Table 2 jcm-11-01960-t002:** Scientific milestones in infectious disease discovery.

Years	Discovery	References
1530	Girolamo Fracastoro expresses his ideas on the origin of syphilis, explaining that this disease is spread by “seeds” distributed by intimate contact	[43]
1683	Anton van Leeuwenhoek observes bacteria under the first microscope	[44]
1701–1714	Giacomo Pilarino and Emmanuel Timoni give the first smallpox inoculations	[45]
1757	Francis Home demonstrates that measles is caused by an infectious agent in the blood of patients	[46]
1796	Edward Jenner develops the process of vaccination for smallpox, the first vaccine for any disease	[45]
1842–1847	Oliver Wendel Holmes describes puerperal fever and Ignaz Semmelweis discovers how to prevent the transmission of puerperal fever	[44]
**1857**	Louis Pasteur identifies germs as a cause of disease	[47]
**1870**	Robert Koch and Louis Pasteur establish the germ theory of disease	[48]
1879	First vaccine developed for chicken cholera by Louis Pasteur	[49]
1881	First vaccine developed for anthrax by Louis Pasteur	[49]
1882	First vaccine developed for rabies by Louis Pasteur	[49]
1882	Koch discovers the Tuberculosis bacillus: *Mycobacterium tuberculosis*	[50]
1890	Emil von Behring discovers antitoxins and develops tetanus and diphtheria vaccines	[49]
1892	Dmitri Ivanovsky shows that sap from a diseased tobacco plant remained infectious to healthy tobacco plants despite having been filtered	[51]
1894	Isolation in culture and microscopic description of causative bacteria	[52,53]
1896/1897	Almroth Wright and Richard Pfeiffer develop the first vaccine for typhoid fever	[54]
1897	Waldemar Haffkine tests on himself the first vaccine developed for bubonic plague	[55]
1897	Paul Ehrlich develops a standardized unit of measure for diphtheria antitoxin that would play an important role in future developmental work on sera and vaccines	[56]
1898	Martinus Beijerinck is convinced that filtrate contains a new form of infectious agent called a virus	[57]
**1913**	Paul Ehrlich develops the first antimicrobial drug, “Salvarsan”, against the bacterium *Treponema pallidum*, the etiological agent of syphilis	[58]
**1918**	Charles Nicolle and Charles Lebailly advance the hypothesis that the causative agent of the “Spanish” flu is a nonfilterable agent of infinitesimal dimensions: possibly a virus	[59,60]
1923	First vaccine developed for diphtheria by Alexander Thomas Glenny	[49]
1924	First vaccine developed for tetanus (tetanus toxoid) by Alexander Thomas Glenny	[49]
1914/1926	First vaccine developed for whooping cough (pertussis) by Leila Denmark	[49]
1927	First vaccine developed for tuberculosis by Albert Calmette and Camille Guérin	[49]
**1928**	Sir Alexander Fleming discovers penicillin	[61]
**1931**	German engineers Ernst Ruska and Max Knoll project the electron microscope	[62]
1931	Live attenuated bacterial vaccine developed and tested	[63,64]
1932	Gerhard Domagk announces that the red dye prontosil is active against streptococcal infections in humans; afterward, Ernest Fourneau, Jacques and Thérèse Tréfouël, Daniel Bovet, and Fedrico Nitti show that the active antibacterial agent is sulfanilamide	[65]
1935	First vaccine developed for yellow fever by Max Theiler	[49]
1937	First vaccine developed for typhus by Rudolf Weigl	[49]
1938	Jonas Salk and Thomas Francis develop the first vaccine against flu viruses	[66]
1940–1947	Large concentrations of blood bacteria correlated with mortality	[67]
1944	Oswald Avery, Colin MacLeod, and Maclyn McCarty report that DNA is the transforming factor in the experiments of Frederick Griffith where an extract of the pathogenic strain of *pneumococcus* could transform a harmless strain into a pathogenic one	[68]
1944	Selman Waksman, Albert Schatz, and Elizabeth Bugie announce the discovery of streptomycin and state that it is active against *Mycobacterium tuberculosis*	[69]
1949	John Franklin Enders, Thomas Weller, and Frederick Robbins grow poliovirus for the first time in cultured human embryo cells	[70]
**1953**	James Watson and Francis Crick describe the structure of DNA	[71]
1955	Jonas Salk develops the first polio vaccine	[49]
1964	First vaccine developed for measles by John Franklin Enders	[49]
1967	Maurice Hilleman develops the first vaccine for mumps virus	[49]
1970	Maurice Hilleman develops the first vaccine for rubella	[49]
1977	First approved vaccine developed for pneumonia	[49]
1978	First approved vaccine developed for meningitis	[49]
1980	Genetic relatedness of *Yersinia pestis* and *Yersinia pseudotuberculosis*	[72]
1981	First approved vaccine developed for hepatitis B	[49]
1983	HIV, the virus that causes AIDS, is identified	[73]
1987	First approval of a drug against AIDS: zidovudine (AZT, ZDV)	[74]
1992	First approved vaccine developed for hepatitis A	[49]
1995	First vaccine developed for varicella (chickenpox) and hepatitis A	[49]
1998	First approved vaccine developed for rotavirus	[49]
2001–2011	DNA sequence of *Yersinia pestis* samples	[75]
2006	First vaccine developed for human papillomavirus and for herpes zoster (shingles)	[49]
2019	Food and Drug Administration (FDA) approves rVSVΔG-ZEBOV-GP Ebola vaccine	[76]
2020	First vaccine against COVID-19	[77]
2022	First drugs for COVID-19 (molnupiravir and Paxlovid or nirmatrelvir + Paxlovid)	[78,79,80]

In bold the most important milestones in science discovery.

In modern history, the most devastating pandemic was the 1918–1919 influenza called the “Spanish” flu (or the “Spanish Lady”) (Table 1). The flu was named “Spanish” since the infection spread in 1918 when the majority of nations were involved in World War I and the newspapers were under censorship rules and only the Spanish reported the pandemic. The first wave of flu appeared in March 1918, at Camp Funston in Kansas (USA), a military training camp. The second, more virulent wave appeared in August. In Europe, it appeared firstly at the seaport of Brest, where the American troops arrived. The confirmed number of casualties is unknown, with an estimated number ranging from 20 to 150 million [101]. The etiological agent of Spanish flu is named H1N1 for the hemagglutinin (H) and neuraminidase (N) proteins. The H1N1 influenza virus is an *orthomyxovirus*. The pandemic was characterized by an uncommonly high mortality rate among healthy young adults (from 15 to 34 years), not observed in either prior or subsequent influenza A epidemics. The case fatality rate was >2.5%, compared to <0.1% in other influenza pandemics [102]. The histopathological analysis of lung tissues of subjects who died in 1918 due to influenza showed acute pulmonary edema and/or hemorrhage with acute bronchiolitis, alveolitis, and bronchopneumonia. Data from DNA sequencing suggest that the 1918 virus consisted of a new virus, and that H1N1 is not a result of recombination of previous virus strains acquiring one or more new genes, as was the case of those causing the 1957 (Asian Flu, H2N2) and 1968 (H3N2) pandemics [103]. In 1918, the biomedical idea of influenza was dictated by Richard Pfeiffer’s 1892 statement that *Bacillus influenzae* was the only etiological cause. Nevertheless, in October 1918, for the first time, two scientists of the Pasteur Institute, Charles Nicolle and Charles Lebailly, advanced the hypothesis that the causative agent of the “Spanish” flu was a nonfilterable agent of infinitesimal dimensions: possibly a virus [60]. Interestingly, in novels and books written during the Spanish flu pandemic period, there is little or no mention of the flu. The majority of deaths likely resulted directly from secondary bacterial pneumonia caused by common upper respiratory tract bacteria [104] of different species that took advantage of the underlying viral infection. As suggested by a study by Anthony Fauci [104], the majority of deaths likely resulted directly from secondary bacterial pneumonia caused by common upper respiratory tract bacteria. According to the above study, the causes of death involved several diverse bacteria, alone or in complex combinations, working with the co-pathogenic properties of the virus itself, possibly linked to viral growth, facility of cell-to-cell spread, cell tropism, or interference with or induction of immune responses. Thus, the large majority of infected subjects in 1918 (>97%) showed a typical, self-limited course of influenza, without any antivirals, antibiotics, or vaccines.

Public health officials imposed multiple interventions as disease containment measures, including the closure of schools, churches, and dancing halls; banning of mass gatherings; mandatory mask wearing; case isolation; and disinfection/hygiene. The primary lesson of the 1918 influenza pandemic was that it is critical to intervene early and that viral spread will be renewed upon relaxation of such measures [105]. 

Starting in the early 1980s, a new and unusual disease began to emerge worldwide. The disease was recognized as pandemic when on June 5, 1981, the US Centers for Disease Control (CDC) of Atlanta received information of unusually high rates of the uncommon diseases *Pneumocystis jirovecii* (then called *Pneumocystis carinii*) pneumonia (PCP) and Kaposi’s sarcoma in young white healthy homosexual men living in Los Angeles. Afterward, in 1982, cases were reported in injection drug users and then in women possibly infected through heterosexual sex. The disease was named acquired immune deficiency syndrome (AIDS) [106]. The first appearance of the disease was in Kinshasa, in the Democratic Republic of Congo, in around 1920, when HIV, the causal agent, crossed species from chimpanzees to humans. In early 1983, a new human retrovirus, initially named lymphadenopathy-associated virus (LAV), was isolated at the Pasteur Institute, Paris, France, from a culture obtained by a lymph node biopsy from a patient with generalized lymphadenopathy. Then, a similar virus was isolated from patients with AIDS, and in 1985, the virus RNA was sequenced. The main receptor for HIV was identified in the CD4 cell surface molecule [73], confirming that HIV causes AIDS. In November 2020, the WHO reported that at the end of 2019, 38.0 million people, 25.7 million of which were in the African Region, were living with HIV [107].

Table 4 summarizes the interventions adopted to prevent the spread of the modern infectious diseases [108,109,110,111,112,113,114,115,116].

## 3. Transmission and Measures to Contain SARS-CoV-2 Spread

After the WHO declared a pandemic [117] on 11th March 2020, most countries, to avoid the pandemic spread and limit the number of casualties, introduced several strict nonpharmaceutical interventions [118], namely (1) improved diagnostic testing and contact tracing; (2) isolation and quarantine for infected people; and (3) measures aimed at reducing mobility and creating social distancing (containment, mitigation, and suppression). 

Most countries decided on the following containment measures: Physical distances > 1.5 m;Wearing masks and gloves;Stay-at-home orders;School and workplace closures and activation of distance learning and smart working;Closure of museums, commercial parks, gyms, and swimming pools;Cancellation of public events;Restrictions on size of crowds;Seat limitation on public transport to ensure the right distance between passengers;Restrictions on internal and international travel;Measurements of body temperature at the entrance of closed areas (<37.5 °C);Ensuring disinfection rules are followed in public areas such as public transport, shopping areas, schools, and universities;Protecting healthcare workers with appropriate personal protection equipment (PPE).

The main objective of these interventions is to reduce the reproduction number (Rt) of the virus. The Rt is defined “as the mean number of secondary cases generated by a typical primary case at time *t* in a population calculated for the whole period over a 5-day moving average” [59]. Thus, Rt is an indicator measuring the transmission of SARS-CoV-2 before and after the interventions.

According to Ecclesiastes 1:9 [קהלת, Qohelet הַשָּֽׁמֶשׁתַּ֥חַת כָּל־חָדָ֖שׁ וְאֵ֥ין Latin Ecclesiastes *Nihil sub sole novum*], “There’s nothing new under the sun.” Table 5 [101,102,103,104,105] reports the containment measures adopted over the millennia to avoid disease spread.

In the past, against some infectious diseases, medicine was ineffective [85] Thucydides in his History of Peloponnesian War (II, vii3-5) wrote “The doctors were unable to cope, since they were treating the disease for the first time and in ignorance: indeed, the more they came into contact with sufferers, the more liable they were to lose their own lives.” Indeed, the only possible way to escape the plague was to avoid any contact with infected persons and contaminated objects. The Italian poet Giovanni Boccaccio (1313–1375), in his book The Decameron (1349–1353), tells the story of ten people, seven women and three men, who entertain themselves with novels while in isolation from the plague of Florence in a villa in the countryside. In the first chapter, Boccaccio describes how the plague struck the city of Florence, how people reacted, and the staggering death toll. Boccaccio, echoing Thucydides, also wrote: “Neither a doctor’s advice nor the strength of medicine could do anything to cure this illness”.

Accordingly, the procedure of obligatory quarantine was introduced as a measure to isolate and separate people, animals, foods, and objects that may have been exposed to a contagious disease. Quarantine is from the Italian “quaranta”, meaning forty. For millennia, contagious diseases were believed to be a divine punishment for sinners. Thus, in the Old Testament, God destroyed the earth with water for 40 days (בראשית: Genesis 7:4); Noah waited for forty days after the tops of mountains were seen after the flood (בראשית: Genesis 8:5–7). Moses was on Mount Sinai for 40 days (Exodus שְׁמֹות 24:18): “Then Moses entered the cloud as he went on up the mountain. And he stayed on the mountain forty days and forty nights”. In the New Testament, Jesus was tempted for 40 days (Matthew 4:2, Mark 1:13, Luke 4:2). There were 40 days between Jesus’ resurrection and ascension (Acts 1:3). Eugenia Tognotti [121] and Gensini et al. [124] reviewed the origin of quarantine from the time of the Bible to nowadays. 

Nevertheless, 40 days may derive from the Pythagorean theory of numbers; according to Pythagoreans, the number 40 was considered to be sacred. Hippocratic teaching in the 5th century BCE established that an acute illness only manifested itself within forty days [124]. Only during the epidemic of 1347–1352 was an organized institutional response to control disease set up. Quarantine was introduced, for the first time, in 1377 by the Rector of the seaport of Ragusa (Dubrovnik, Croatia), and the first stable plague hospital (lazaretto or quarantine station) was built by the Republic of Venice in 1423 on the island of Santa Maria di Nazareth [124]. The term lazaretto, usually referred to as Nazarethum or Lazarethum, is related to Lazarus, who was brought back to life by Jesus (John 11:1–45) and/or to the Order of Saint Lazarus of Jerusalem, a Catholic military order founded by crusaders around 1119 at a leper hospital in Jerusalem, as a hospital and military order of chivalry [125]. The Venetian system became a model for other European countries: in 1467, Genoa adopted the Venetian system, and in 1476, in Marseille, France, a hospital for persons with leprosy was converted into a lazaretto [126]. Afterward, quarantine became the foundation of a coordinated disease-control strategy that included different measures such as isolation, sanitary cordons, bills of health issued to ships (certification assuring the absence of disease), sanification (i.e., fumigation), and disinfection. Girolamo Fracastoro, Latin Hieronymus Fracastorius (1478–1553, Verona), physician, poet, astronomer, and geologist, was the first to propose, in 1546, a scientific germ theory of disease. In his book “On Contagion and Contagious Diseases”, he affirmed that each disease is caused by a different type of rapidly multiplying minute body and that these bodies are transferred from the infector to the infected in three ways: by direct contact; by carriers such as soiled clothing and linen; and through the air.

## 4. Boosting the Immune Response: How Vaccines Changed the Scenario

In the millennial history of mankind, vaccination is a relatively young intervention of primary prevention. For about 200 years, vaccination strategies have had a profound effect in shaping the natural history of infectious diseases. Smallpox eradication represents the most impressive success of a vaccination strategy. As discussed above, smallpox represented a dreadful menace throughout the centuries [127]. It was common knowledge that smallpox survivors acquired immunity to the disease, so the practice of variolation, consisting in having healthy individuals inhale dust from smallpox lesions, become common in Europe and in North America. At the end of the 18th century, there were anecdotes regarding immunity to smallpox in people previously infected with cowpox, a zoonotic pathogen [128]. In 1798, Edward Jenner published his first observations on the benefits of inoculating biological material from cowpox lesions in humans, to protect from smallpox, and for the first time, the term “vaccination” (from Latin vacca, English cow) was used [129]. Initially, vaccination was perpetuated by transferring fluids from individual to individual, but this practice reduced the strength and the duration of protection. The next step was to deliberately infect cows to mass-produce sufficient material (“lymph”) for vaccination [127]. However, this practice led to an increase in the frequency of transmission of secondary infections, including syphilis. This issue was resolved after the observation of bacterial inactivation by glycerin made by Robert Koch [130], so lymph was treated with glycerin before inoculation. At the end of the 19th century, Louis Pasteur made observations that strongly enhanced the development of vaccines. Studying chicken cholera, he noticed that chickens inoculated with cultures left out over a prolonged period were protected from subsequent infection with fresh material, suggesting the existence of protective immunity induced by the inoculation of “aged” material. These observations would lead Louis Pasteur to the production of antirabic vaccination, using material from an infected dog’s brain, exposed to dry air [131]. 

In the meantime, studies on the immune system contributed to unravel the mechanisms of host defense from pathogens. In particular, studies by Elie Metchnikoff in 1884 [132] introduced the concept of cellular immunity, and Paul Ehrlich published his theory of receptor of immunity in 1897, paving the way for the development of antitoxins against pathogens such as diphtheria. At the end of the 19th century, five human vaccines were in use: two live virus vaccines (smallpox and rabies) and three dead bacterial vaccines (typhoid, cholera, and plague.).

The first half of the 20th century saw the development of passive immunization, with the production of antitoxins for diphtheria and tetanus. At the same time, new vaccines against tuberculosis, bacillus Calmette-Guérin (BCG), yellow fever, typhus, influenza A, and pertussis were developed, and the first combination vaccine, against diphtheria, tetanus, and pertussis was produced in 1948. Progresses in cell culture lead the way to the techniques of virus attenuation through passages on tissues and cellular monolayers, thanks to the studies of Hugh and Mary Maitland in 1928 and Ernest William Goodpasture in 1931, who first used the chorioallantoic membranes of a fertile hen’s egg as a culture medium for sterile passage of viruses. In the 1900s, poliomyelitis (caused by the poliovirus) represented another threat to public health. The virus spreads from person to person and can invade an infected person’s brain and spinal cord, causing paralysis. Better hygiene conditions led to an increase in the age of infected children, which in the previous centuries were breastfed, protected by maternal antibodies. The older age of infected children led to frequent polio outbreaks. The first effective antipolio vaccine was a formaldehyde-inactivated (or “killed”) PV vaccine (IPV) developed by Jonas Salk in 1955. A second vaccine which was demonstrated to be both safe and effective was the oral (or “live”) PV vaccine (OPV) was developed by Albert Sabin in 1963 [133]. Jonas Salk and Albert Sabin decided not to patent their vaccines and therefore sacrificed billions of dollars in potential royalties, approximately USD 500 million for Salk and approximately USD 1.2 billion for Sabin. Nowadays, thanks to the vaccines, the virus remains endemic only in Afghanistan and Pakistan.

In more recent years, research started to focus on multiple vaccines, starting from live viruses attenuated by multiple passages on cultured cells, such as the vaccine against measles, mumps, and rubella. Exploiting the newly available techniques of molecular biology, newly designed vaccines started to be produced. Japanese researchers developed an acellular pertussis vaccine based on two of the main protective antigens of *Bordetella pertussis*. Research on polysaccharide vaccines led to the development of new vaccines against pneumococcus, meningococcus, and *Haemophilus Influenzae* type B. Recombinant DNA technology and the possibility to produce recombinant protein in vitro paved the way for the release of the anti-hepatitis B vaccine. Under the urgent need to battle COVID-19, different SARS-CoV-2 vaccines, including the inactivated virus vaccine, nucleic acid vaccine, adenovirus vector vaccine, and viral subunit vaccine, have been developed [134]. In the history of vaccines, COVID-19 vaccines are unique for the extraordinary rapidity of their production. In recent years, mRNA vaccines have started to attract great attention thanks to their potential to:(1)Speed up vaccine development;(2)Simplify vaccine production, scale-up, and quality control;(3)Be produced and scaled up in a predictable and consistent fashion regardless of the antigen;(4)Have improved safety and efficacy;(5)Challenge diseases impossible to prevent with other approaches;(6)Enable precise antigen design;(7)Generate proteins with a “native-like” presentation;(8)Express proteins stabilized in a more immunogenic conformation or expose key antigenic sites;(9)Deliver multiple mRNAs to the same cell;(10)Allow the generation of multiprotein complexes or protein antigens from different pathogens, thus creating a single vaccine against several targets.

Moreover, mRNA is characterized as:
(11)Being noninfectious;(12)Being nonintegrating;(13)Being degradable by normal cellular processes soon after injection;(14)Decreasing the risk of toxicity and long-term side effects;(15)Not inducing vector-specific immunity;(16)Not competing with pre-existing or newly raised vector immunity that could interfere with subsequent vaccinations.

An mRNA vaccine is based on the principle that mRNA is an intermediate messenger to be translated to an antigen after the delivery into host cells via various routes. The mRNA is synthesized in the laboratory by transcribing a DNA template of the genetic sequence encoding the immunogen. In the case of SARS-CoV-2, the spike (S) protein is identified as the immunodominant antigen of the virus. The most important problem is that mRNA is unstable, easily recognized by the immune system, and rapidly degraded by nucleases after entering the body. mRNA vaccines do not enter the nucleus but need to pass through the cell membrane, a negatively charged phospholipid bilayer, to enter the cytoplasm and then be translated into the target protein. Different delivery systems for mRNA vaccines, such as viral and nonviral vector delivery systems, may be utilized. Vectors based upon lipids or lipid-like compounds are the most common nonviral gene carriers.

## 5. Conclusions

The deep knowledge of the history of the “plagues” that have struck humanity is not only precious in understanding the long-term sociological and demographic changes but in better understanding the evolution of infectious diseases over the centuries. Reading all the literary works carefully, we can recognize our present journeys under the COVID-19 pandemic, including the risks of inappropriate responses. Confinement measures such as social distancing and/or quarantine still remain the most efficient measure to contain the spread of the virus. As in the past, an “infodemia” is present, generating chaos and fear among the population. Moreover, technical knowledge on agents, hosts, and the environment alone, although essential, is not enough. 

The COVID-19 pandemic is an example of how diseases unknown to medical science and to human immune systems may develop and spread quickly in a highly connected world. Even it is not possible to avoid every risk, there are ways to reduce or mitigate the chances of a future pandemic, such as investing in research and preparation, funding and implementing vaccine programs, and strengthening health systems. Along with vaccines and specific therapies, the best course of action in facing new pandemics remains social distancing, the practice of good hygiene, and the use of quarantine.


*“And Darkness and Decay and the Red Death held illimitable dominion over all.”*
(“The Mask of the Red Death: A Fantasy” E. A. Poe, 1842) 

## Figures and Tables

**Figure 1 jcm-11-01960-f001:**
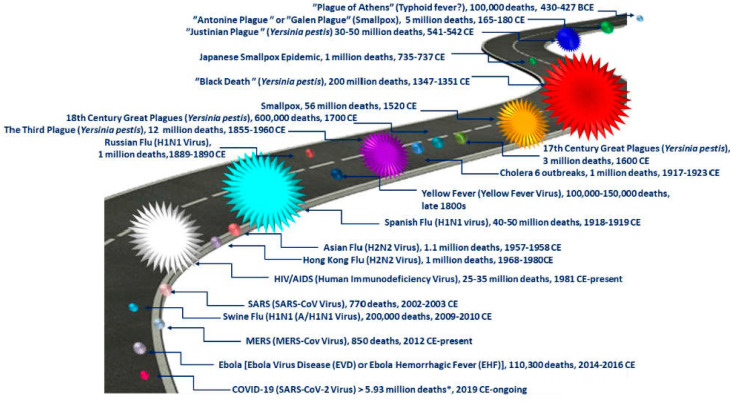
Emergence of the most important pandemics in the world until today with a possible estimation of the number of casualties. * the number is not definitive since the pandemic is ongoing. Adapted from [31].

**Figure 2 jcm-11-01960-f002:**
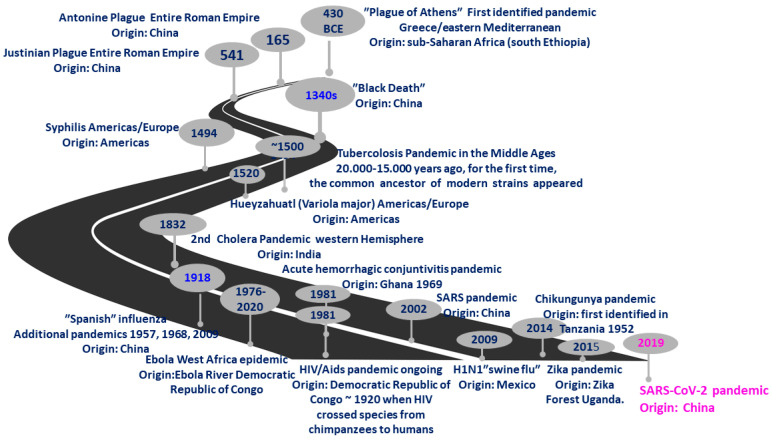
First place of appearance of pandemics.

**Table 1 jcm-11-01960-t001:** Plague in the human literary documents.

Years	Infectious Disease	Literary Documents	Origin	Possible Pathogen	Peer-Reviewed Source
1500 BCE	Different diseases	Ebers Papyrus	Egipt	Unknown	[10]
1335 BCE	Hittite plague	The Amarna Letter EA 96; The Amarna Letter EA 137; The Amarna Letter EA 224.	Mesopotamia via Canaanite harbors	*Francisella tularensis*	[11]
		The story of the plague is handed down in literature in the Hittite plague prayers (KUB XIV 10 and KUB XXVI 86)			
1190 or 1141 BCE	Plague of the Philistines, also known as the Plague of Ashdod	Bible, Book of Samuel according to 1 Samuel 5:9	Ashdod (now Israel)	*Y. pestis, Francisella tularensis*, smallpox, or variola	[12]
800–781 BCE	Plague of the Achaean soldiers’ camp	Homer’s Iliad Book 1:50–56	Troy or Ilion located at Hisarlik in present-day Turkey	Equine encephalomyelitis?	[13]
463 to 462 BCE	Rome when Lucius Aebutius and Publius Servilius were elected consuls	Quintus Livius Titus, popularly known as Livy, in his Early History of Rome	Rome, Italy	Unknown	[14]
430–420 BCE	Plague of Thebes	Oedipus Rex by Sophocles (original Greek title Oιδίπους τύραννος)	Thebes, Greece	*Leishmania* spp., *Leptospira* spp., *Brucella abortus, Orthopoxviridae*, and *Francisella tularensis* in cattle.	[15]
				Humans could have been affected by a different pathogen such as *Salmonella enterica* serovar *Typhi*	
430–429 BCE	Plague of Athens	*History of the Peloponnesian War* by Thucydides	Ethiopia, then all the Mediterranean	*Salmonella enterica* serovar *Typhi*	[16,17]
430–429 BCE	Plague of Athens	Titus Lucretius Carus 99–55 BCE De Rerum Natura with the Plague of Athens	Athens, Greece	Unknown	[18]
433–432 BCE	Rome when Gaius Julius and Lucius Verginius were consuls	Quintus Livius Titus, popularly known as Livy, in his Early History of Rome Ab Urbe Condita 4.20–21, 4.25.3–4, and 4.30.8–10)	Rome, Italy	Anthrax?	[14]
396 BCE	Plague at Syracuse	*Punica* by Tiberius Catius Asconius Silius Italicus	Syracuse, Italy	Typhoid or smallpox	[19]
Mythological plague	70–19 BCE	Publius Vergilius Maro Third book of Georgics	Unknown	Unknown	[18]
Mythological plague	Plague at Aegina	Publius Ovidius Naso 43 BCE–18 CE Metamorphoses, Book 7 523–613	Unknown	Unknown	[18]
Mythological plague	Lucius Annaeus Seneca Between 8 and 1 BCE–65 CE	Oedipus is a fabula crepidata (Roman tragic play with Greek subject)	Unknown	Unknown	[18]
293 BCE	Plague of Rome	The construction of a hospital on Tiber Island in honor of Asclepius, the God of Medicine	Rome, Italy	Unknown	[14]
165 CE, first wave	The Antonine Plague, Marcus Aurelius Antoninus (161–180 CE)	Plague of Galen, Methodus Medendi	Rome, Italy	Smallpox (*Variola major* and *Variola minor*)	[20,21]
251–266 CE second wave	Cyprian Plague	Documented by Saint Cyprian, the Assyrian	Ethiopia	Small-pox	[21]
541–543 CE and subsequent outbreaks 750–1000 CE	Justinian Plague Justinian I (r. 527–565 CE)	Symptoms described by Procopius of Cesarea (500–565 CE) in his History of the Wars, Book II, are very similar to those later reported during the second pandemic of the 14th to 17th centuries (the Black Death)	Pelusium, near Suez in Egypt	*Y. pestis*, distinct from the lineage that caused the second pandemic (including the Black Death) 800 years later in human populations	[22]
590 CE	Roman Plague	The plague was described by the bishop and chronicler Gregory of Tours and later chronicler Paul the Deacon	Rome, Italy	*Y. pestis*	[14]
1347–1351	Black Death		China and Inner Asia	*Y. pestis*	
	Thomas Walsingham—Norfolk, 1422	Thomas Walsingham “Towns once packed with people were emptied of their inhabitants, and the plague spread so thickly that the living were hardly able to bury the dead”		*Y. pestis*	[23]
	Gabriele de Mussi (1280–1356)	Gabriele de Mussi Historia de Morbo		*Y. pestis*	[24]
	Francesco Petrarch (1304–1374)	Francesco Petrarch chronicles the plague at Parma and bewails the magnitude of the destruction that seems to threaten the very existence of the human race		*Y. pestis*	[23]
	Agnolo di Tura (14th century)	Agnolo di Tura, in his Crunica Senese, gives a shocking account of the raging plague in Siena		*Y. pestis*	[25]
	Simon de Covino (1320–1367)	At Marseilles, the Black Death entered France, and Simon de Covino, the French poet–physician, describes the symptoms of the disease in Latin hexameter verse		*Y. pestis*	[25]
	Giovanni Boccaccio 1313–1375	Boccaccio acknowledges the Florentine plague of 1348 to be the prime mover of The Decameron, and in the introduction to that work he says “between March and the following July”		*Y. pestis*	[26]
	Matteo Villani 1275–1363	Matteo Villani attributes to the Black Death the social, moral, and political anarchy that were rampant in Florence in succeeding years		*Y. pestis*	[27]
The second plague pandemic, caused by *Yersinia pestis*, devastated Europe and the nearby regions between the 14th and 18th centuries 1360–1363; 1374; 1400; 1438–1439; 1456–1457; 1464–1466; 1481–1485; 1500–1503; 1518–1531; 1544–1548; 1563–1566; 1573–1588; 1596–1599; 1602–1611; 1623–1640; 1644–1654; 1664–1667; 1679	Second outbreak of Black Death	Giovanni Baldinucci’s diary		*Y. pestis*	[28]
	Italian Plague of 1629–1631 or the Great Plague of Milan and the Great Plague of Seville (1647–1652),	Francesco Rondinelli			
	the Great Plague of London (1665–1666),	Plague of Florence 1632–1633			
	and the Great Plague of Vienna (1679)	Alessandro Manzoni’s The Betrothed			
1720–today	Third outbreak of Black Death. There is some controversy over the identity of the disease, but in its virulent form, it was responsible for the Great Plague of Marseille in 1720–1722, the Great Plague of 1738 (which hit Eastern Europe), and the Russian plague of 1770–1772		Yunnan Province, Southwest China	*Y. pestis*	[23]

**Table 3 jcm-11-01960-t003:** Infectious diseases studied using molecular techniques including metagenomics.

Time Frame of Disease Appearance	Archaeological Discovery and Pathogen Identification	Method of Retrieval	References
4900 BCE	Different individuals at Fralsegarden in Gokhem parish, Falbygden, western Sweden. Identification of the presence of *Y. pestis* DNA	Genotyping of *Y. pestis* strain, phylogenetic and molecular clock analyses, heatmaps and functional classification of variants, and admixture analyses of human genomes	[8]
4800 to 3700	563 tooth and bone samples from Russia, Hungary, Croatia, Lithuania, Estonia	Shotgun screening sequencing, in silico screening, deep shotgun sequencing, Y. pestis in-solution capture, genome reconstruction and authentication, individual sample treatment due to laboratory preparation and sequencing strategies, SNP calling, heterozygosity and phylogenetic analysis, dating analysis, SNP effect analysis, virulence factor analysis, and indel analysis	[88]
	Latvia, and Germany.		
	Identification of the presence of extinct clade in the *Y. pestis* phylogeny		
3250 BCE to 700 CE	Different individuals from Abusir el-Meleq, located in Middle Egypt.	Metagenomic screening and SNP typing	[89]
	Identification of *Mycobacterium leprae* strain and a 2000-year-old human hepatitis B virus		
3500 BCE to 300 CE	Mummies from Upper and Lower Egypt.	PCR amplification and sequencing	[90,91,92]
	Identification of Falciparum malaria (*Plasmodium Falciparum*) and human tuberculosis (*Mycobacterium tuberculosis*)		
430 BCE	Dental pulp in Kerameikos mass burial of putative victims of the plague.	PCR amplification and sequencing	[32]
	Identification of *Salmonella enterica* serovar *Typhi*		
541–543 CE	Radiocarbon dating of individuals to 533 AD (plus or minus 98 years) from the Aschheim-Bajuwarenring, Bavaria, Germany.	PCR amplification and sequencing reconstructed draft genomes of the infectious *Y. pestis* strains, comparing them with a database of genomes from 131 *Y. pestis* strains from the second and third pandemics, and constructing a maximum likelihood phylogenetic tree.	[22]
	Identification of *Y. pestis*		
13th and beginning of the 14th century to 19th century	Different subjects from different places in Europe. Identification of *Y. pestis*	PCR and sequencing SNP calling and evaluation	[93,94,95,96]
1630–1632	Different subjects from Imola Northern Italy. Identification of *Y. pestis*	Genomic and metagenomic analysis of sequencing data	[97]
16th century	Italian mummy from the 16th century.	Genomic and metagenomic analysis of sequencing data	[98]
	Identification of intact smallpox virus particles		
1580–1630 VARV	Different subjects from Vilnius, Lithuania.	PCR and sequencing	[99,100]
	Identification of smallpox		

**Table 4 jcm-11-01960-t004:** Modern infectious disease prevention and intervention.

Pandemic	Years	Source	Main Actions	Current Prevention	References
Yellow fever	1900	Mosquito	Travel limitation in areas affected by yellow fever. Yellow fever control strategies include insecticide spraying, larval control including larvicide spraying, and bacterial toxins	Vaccine valid for 10 years	[108,109]
SARS-CoV	2002–2004	Bats, palm civets	During the viral incubation period (4–5 days), which is important for the prevention and control of the disease, there are no clinical symptoms	One of the treatment modalities used to reduce the replication of the virus and its spread is passive immunization with monoclonal antibodies	[110]
MERS-CoV	2012	Dromedary camels	Prevention includes washing hands often, cleaning surfaces regularly with an alcohol-based cleaner,	Interferons (IFNs). Different vaccines targeting SARS-CoV and MERS-CoV have been developed and tested in preclinical models. However, only a few of them have gone into clinical trials, and none of them have been approved by the FDA.	
			covering mouth and nose with a tissue when coughing or sneezing	The different vaccines include protein subunit vaccines (RBD-based vaccine), virus-like particle vaccines, DNA vaccines, viral vector vaccines, inactivated vaccines, and live attenuated vaccines	
A/H1N1	2009	Pigs	Reducing the risk of human-to-human transmission: isolation and quarantine of infected patients. Use of the surgical template. Hand hygiene is the most important measure to reduce the risk of transmission. Hands should be washed frequently with soap and water, alcohol-based cloths, or antiseptic. Cleaning of contaminated surfaces or equipment should be performed with phenolic disinfectants, ammonia compounds, or alcohol	Antiviral drugs, including adamantanes (amantadine, rimantadine) and neuraminidase inhibitors (zanamivir, oseltamivir, peramivir, and laninamivir) are used to treat cases of influenza, even if they have side effects; antibiotics for the treatment or prevention of secondary bacterial pneumonia; parenteral nutrition; oxygen therapy or ventilatory support and vasopressors for shock. Vaccines: for subjects between the ages of 3 and 77. The immunization schedule consisted of two vaccinations, 21 days apart	[111,112]
Ebola	2013–2016	Bats, NHPs, and small terrestrial mammals	Reducing the risk of wildlife-to-human: avoiding contact with infected fruit bats or monkeys/apes and the consumption of their raw meat.	Combination of three mono-clonal antibodies directed against the envelope glycoprotein (GP) of EBOV, liposomal-formulated interfering RNA, and inhibitors of RNA polymerase.	[113,114]
			Reducing the risk of human-to-human transmission: avoiding direct or close contact with people with Ebola symptoms, particularly with their bodily fluids. Gloves and appropriate personal protective equipment should be worn when taking care of ill patients at home. Regular hand washing is required after visiting patients in hospital, as well as after taking care of patients at home	Two main vaccines have proved efficacious in preventing Ebola infection. Both vaccines express GP as the single EBOV component and are virally vectored in chimpanzee adenovirus and vesicular stomatitis virus (rVSV), respectively	
			Outbreak containment measure: prompt and safe burial of the dead, identifying people who may have been in contact with someone infected with Ebola, monitoring the health of contacts for 21 days, the importance of separating the healthy from the sick to prevent further spread, the importance of good hygiene and maintaining a clean environment		
SARS-CoV-2	2019	Probably from bats	Several practices are recommended with the aim to limit further transmission; they include handwashing, hand disinfection, wearing of face masks and gloves, disinfection of surfaces, and physical distance	Conservative fluid therapy and broad-spectrum antibiotics are given to patients as a protective measure to avoid opportunistic bacterial infections. However, ventilator support for respiration is provided to patients under extreme conditions.	[115,116]
				Numerous FDA-approved antiviral drugs, plasma therapy, vaccines (live attenuated vaccine (LAV), inactivated virus, subunit vaccines, monoclonal antibody vaccine, virus vectors, protein vaccines, and DNA/RNA-based vaccines), and immunotherapies	

**Table 5 jcm-11-01960-t005:** Containment measures to avoid disease spread over the millennia.

Years	Source	Measurements	References
430–428 BCE	Plague of Athens	The containment strategies used included the application of purifications and incantations and the enforcement of abstinence from baths and many food items then considered noxious to diseased people	[119]
541–755	Plague of Justinian	The containment strategies included unspecified traditional public health measures and quarantine	[119]
	Giovanni Boccaccio (1313–1375), in his book *The Decameron*	Written in Tuscan vernacular (Italian). The book is a collection of short stories told by a group of seven young women and three young men sheltering in a villa just outside Florence to escape the black death that afflicted that city. Boccaccio probably conceived his masterpiece of classical Italian Renaissance prose after the plague epidemic of 1348, which came to a standstill by 1353	
1377, 1397	“De ordinibus contra eos qui veniunt de loc ispestiferis anno 1397 factis”	Quarantine was first introduced in Dubrovnik on Croatia’s Dalmatian Coast	[120]
	Orders made against those who come to the place of the pestiferous in 1397	The Great Council of Ragusa specified again the 30-day duration of quarantine and determined the place	
1423, 1448	Venetian Senate	First permanent plague hospital (lazaretto) was opened by the Republic of Venice in 1423 on the small island of Santa Maria di Nazareth.Prolonged the waiting period to 40 days, thus giving birth to the term “quarantine”	[121]
1467		Genoa adopted the Venetian system	[119]
1476		In Marseille, France, a hospital for persons with leprosy was converted into a lazaretto	[121]
1480	Marsilio Ficino, “Consilio contro la pestilentia”	When you converse, stay away from your partner at least two arms, and in the open place, and when it is suspicious, let us stay at least six fathoms longer, and out in the open, and let the wind not be reversed by him	
1589	Viceroy of Peru	Lima physicians advised the use of quarantine among all native communities to prevent further spread of the disease	[119]
1663		English quarantine regulations provided for the confinement (in the Thames estuary) of ships with suspected plague-infected passengers or crew	[121]
1665	A journal of the plague year by Daniel Defoe	It was a rule with those who had thus two houses in their keeping or care, that if anybody was taken sick in a family, before the master of the family let the examiners or any other officer know of it, he immediately would send all the rest of his family, whether children or servants, as it fell out to be, to such other house which he had so in charge, and then giving notice of the sick person to the examiner, have a nurse or nurses appointed, and have another person to be shut up in the house with them (which many for money would do), so to take charge of the house in case the person should die	[122]
1688 and 1691		Quarantine to control yellow fever which first appeared in New York and Boston	[121]
1796		United States introduced quarantine legislation in port cities threatened by yellow fever from the West Indies	[123]
1799		In the harbor of Philadelphia, the first quarantine station was built after a previous yellow fever outbreak in 1793	[123]
1878		Release of the National Quarantine Act, which shifted quarantine power from single states to the federal government	[123]
1944		The federal government quarantine authority was set up	[123]

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
