# Peer review of "Preventive Measures against Pandemics from the Beginning of Civilization to Nowadays—How Everything Has Remained the Same over the Millennia"

_jcm, 2022, doi:10.3390/jcm11071960_

Round 1
Reviewer 1 Report
This paper is an interesting historical tour of pandemics, social responses to the same, and how recent innovations in vaccines have influenced social responses to pandemics. The paper is well structured and documents a great number of instances of pandemics over history. I would only recommend a quick review of English mistakes.
Author Response
Reviewer #1: Comments and Suggestions for Authors
This paper is an interesting historical tour of pandemics, social responses to the same, and how recent innovations in vaccines have influenced social responses to pandemics. The paper is well structured and documents a great number of instances of pandemics over history. I would only recommend a quick review of English mistakes.
RE: We corrected all English mistakes

Reviewer 2 Report
I recommend rewriting after including a scientific account of the history of infectious diseases and citing peer-reviewed articles as references. The authors relied on directly citing religious and mythical scripts as references, which is not the place for a clinical journal review article.
The attached PDF contains specific comments for the authors to improve their manuscript. Below are more recommendations:
- All tables need rewriting. Specific comments to help authors with this task are written in the PDF.
- For Table 2, authors must keep the review article's main aim clear to the reader. The aim as written in title and abstract: "the evolution of the preventive measures against pandemics over the history of infectious diseases".
- Figures 1 and 2 must be improved and resolution must also be improved to be legible to read.

Round 2
Reviewer 2 Report
I recommend that the authors use paleomicrobiology literature to rewrite the following parts of the manuscript after removing non-scientific parts and speculations from tables:
revised Table 1 and revised Table 4
Add an account of history of infectious diseases studied using molecular techniques including metagenomics with peer-reviewed scientific evidence to back those findings. To mention a few of such articles, authors might find it useful to read the following:
Neukamm, Judith et al. “2000-year-old pathogen genomes reconstructed from metagenomic analysis of Egyptian mummified individuals.” BMC biology vol. 18,1 108. 28 Aug. 2020, doi:10.1186/s12915-020-00839-8
Lalremruata A, Ball M, Bianucci R, Welte B, Nerlich AG, Kun JF, Pusch CM. Molecular identification of falciparum malaria and human tuberculosis co-infections in mummies from the Fayum depression (Lower Egypt). PLoS One. 2013;8(4):e60307. doi: 10.1371/journal.pone.0060307. Epub 2013 Apr 2. PMID: 23565222; PMCID: PMC3614933.
Nerlich, Andreas G et al. “Plasmodium falciparum in ancient Egypt.” Emerging infectious diseases vol. 14,8 (2008): 1317-9. doi:10.3201/eid1408.080235
Zink, A R et al. “Molecular study on human tuberculosis in three geographically distinct and time delineated populations from ancient Egypt.” Epidemiology and infection vol. 130,2 (2003): 239-49. doi:10.1017/s0950268802008257
Zink, Albert R et al. “Molecular analysis of ancient microbial infections.” FEMS microbiology letters vol. 213,2 (2002): 141-7. doi:10.1111/j.1574-6968.2002.tb11298.x
Moreover, there are reviews on the subject:
Review by Donoghue, Helen D. “Paleomicrobiology of Human Tuberculosis.” Microbiology spectrum vol. 4,4 (2016): 10.1128/microbiolspec.PoH-0003-2014. doi:10.1128/microbiolspec.PoH-0003-2014
A great review by Donoghue, Helen D. “Insights into ancient leprosy and tuberculosis using metagenomics.” Trends in microbiology vol. 21,9 (2013): 448-50. doi:10.1016/j.tim.2013.07.007
Review by Brier, Bob. “Infectious diseases in ancient Egypt.” Infectious disease clinics of North America vol. 18,1 (2004): 17-27. doi:10.1016/S0891-5520(03)00097-7
The authors are still relying on non peer-reviewed sources as references. Instead of authors' interpretation of the Bible Old Testament or Torah and citing it, authors should use the scientific interpretations of such scripts as studied by scientific historians. For example Egyptian "Ebers Papyrus" and other books of medicine of ancient Egypt, there is a book by John F. Nunn called "Ancient Egyptian Medicine" published 2002 by University of Oklahoma Press.
Other changes:
Figure 2 title needs rewriting
reference list is missing information of some references e.g. ref. 33
